# Spatial Prioritization of Ecosystem Services for Land Conservation: The Case Study of Central Italy

**Alessandro Sebastiani** [1,*] and **Silvano Fares** [2,3]

1    Council for Agricultural Research and Economics (CREA), Research Centre for Forestry and Wood (FL), 00166 Rome, Italy
2    National Research Council of Italy, Institute of BioEconomy, Via dei Taurini 19, 00185 Rome, Italy
3    National Research Council of Italy, Institute for Agriculture and Forestry Systems in the Mediterranean, P.le Enrico Fermi 1-Loc, Porto del Granatello, 80055 Portici, Italy
*    Correspondence: alessandro.sebastiani@crea.gov.it

**Abstract:** Ecosystem services delivered by natural ecosystems are increasingly important for climate change adaptation and mitigation and play a huge role in biodiversity conservation. For this reason, the EU has the ambitious goal of protecting at least 30% of land by 2030. Member states are called to improve and expand the network of protected areas within the next few years; to do so, scientific studies aimed at identifying areas with high ecological value, as well as at defining best management practices, are highly needed. In this study, we used the InVEST suite of models to spatially assess three regulating ecosystem services, that is, carbon storage, seasonal water yield, and urban flood risk mitigation in three administrative regions of central Italy. Using overlay analysis, we found areas with the highest delivery in each of the considered ESs; based on these findings, we eventually proposed four new protected areas, which combine for 888 km$^2$, that is, 2.73% of the study area. Interestingly, each of the newly proposed protected areas has somehow been discussed and hypothesized by stakeholders, but only one is presumably going to be part of the national network of protected areas within the next years. Hopefully, by prioritizing areas according to the production of ecosystem services, this study can be intended as a step towards the systematic inclusion of ecosystem services studies for enhancing the network of areas under national protection schemes and achieving the goal of protecting at least 30% of land in Europe by 2030.

**Keywords:** protected areas; regulating ecosystem services; spatial analysis; land conservation

## 1. Introduction

Protecting natural capital has become increasingly important over the last years in order to mitigate environmental and social issues related to climate change [1,2]. Indeed, through the delivery of ecosystem services (ESs), nature can sequester and store $CO_2$, remove air pollutants from the atmosphere, prevent floods caused by extreme precipitation events, lower the air temperature during heatwaves, host remarkable biodiversity, and provide space for recreational and cultural activities [3–7]. ESs may also boost social development by creating economic opportunities such as ecotourism [8].

The European Commission (EC) has widely acknowledged the role of nature in improving human health and well-being, recognizing that nature-based solutions (NBSs), which include the protection of natural ecosystems, are essential for climate change adaptation [9]. In the EU Biodiversity Strategy for 2030, it is stated that at least 30% of land should be protected by 2030, one-third of which under strict protection, which means a minimum of an extra 4% compared to today's levels. Plus, efforts should be made to ensure high connectivity amongst protected areas, to create a coherent trans-European Nature Network [9]. The EC has also recently proposed the Nature Restoration Law, a pioneering proposal that aims at preventing ecosystems' collapse and the negative impacts exerted by climate change through numerous restoration measures. Protected areas are indeed

fundamental for biodiversity conservation, protection of cultural heritage, promotion of research and environmental education, sustainable economic development, and ecosystem services preservation [10]. In this framework, EU member states are called to extend the network of protected areas; however, this is a complex process, and a multitude of aspects, including the environmental, social, and economic needs of the population, must be addressed [11]. For this reason, the extension of the network of protected areas must be supported by science through tailored studies, with reliable and easy-transferrable results that allow for informed decision-making [12]. As a mitigation strategy for pollution and climate extremes, the Italian government with its National Recovery and Resilience Plan (PNRR), recently opened a call for EUR 300 million for planting over 6 million trees in 14 metropolitan cities, including Rome, Milan, and Naples. Such unprecedented efforts call into play the need to identify new areas for afforestation/reforestation initiatives [13].

Due to its outstanding climatic, biogeographic, and genetic diversity, Italy is considered one of the most important biodiversity and ES hotspots in Europe [14]; as a consequence, the network of protected areas is well settled. Nationally protected areas of Italy are listed in the Official List Of Protected Areas (EUAP), established according to law 394/91. EUAP counts 871 protected areas, including 24 national parks and 134 regional parks, and covers about 11% of the national terrestrial territory, which is, about two times greater than the European average [15]. Italy also hosts more than 2600 Natura2000 sites, which are instead established under Community Directives Habitat (1992) and Birds (1979). It is worth reminding one that EUAP areas and Nautura2000 sites are not necessarily overlapped; indeed, in Italy, about half of Natura2000 sites fall outside the EUAP network. Overall, the combination of EUAP and Natura2000 sites covers about 20% of the national land; therefore, despite being far ahead of many other European member states, the goal of protecting at least 30% of land by 2030 will soon be out of reach for Italy if measures are not taken urgently, either by enlarging the Natura2000 network or setting new protected areas under national protection schemes [16].

This study aims at providing new, scientifically sound, and spatially-explicit information for the enhancement of the network of protected areas in central Italy. Specifically, our goal is to reinforce the role of the ES-based perspective, which has to date been little considered. To do so, we spatially assessed three regulating ESs in three administrative regions of central Italy, that is, Abruzzo, Lazio, and Molise, using the InVEST (integrated valuation of ecosystem services and tradeoffs) suite of models; the considered ESs were carbon storage, seasonal water yield, and urban flood risk mitigation. Using overlay analysis, we identified priority areas (PAs) for the delivery of ecosystem services, and eventually identified four new proposed protected areas (PPAs) that would substantially enhance the protection of ESs. Interestingly, (i) the current EUAP network only covers 31% of the PAs, and (ii) each one of the PPAs has been yet hypothesized and discussed by associations and stakeholders; however, except for one out of the four, institutions have shown a certain disinterest and the discussion around the establishment of these new protected areas has stalled over the last years.

Hopefully, this study can be intended as a further step towards the systematic inclusion of ESs in the land planning and conservation process.

## 2. Materials and Methods

### 2.1. Study Area

The study area includes Lazio, Abruzzo, and Molise, three administrative regions in central Italy (Figure 1). It lies over 32,476 km$^2$ and is populated by more than 7 million people; the most important city is Rome. Central Italy is characterized by a remarkable climatic and biogeographic diversity, which includes plains, hills, coastal areas, and the central Apennines, which are part of the Apennine mountains, a mountain system extended for about 1200 km along the length of Italy. Central Italy's altitude spans from 0 to about 2900 m asl (Figure 1c). The central Apennines are featured by extended forest ecosystems, including beech woods and deciduous oak woods, and dry grasslands above the natural

tree line, which are at approximately 1900 m asl [17]. The EUAP network is already
well developed, as there are numerous national and regional parks such as Abruzzo,
Lazio and Molise national parks, Majella national park, Gran Sasso and Monti della Laga
national parks, Simbruini regional park, and Sirente-Velino regional park, which cover
about 5200 km$^2$ combined, that equates to about 16% of the study area. Both flora and fauna
are extremely rich: for the Majella national park alone, a study reported 2286 specific and
sub-specific floristic taxa, and 15 exclusive endemics [18] (Conti et al., 2019); as for the fauna,
the central Apennine hosts more than 70 species listed in the habitat directive, including
the critically endangered species *Ursus arctos marsicanus* and the vulnerable species *Canis
lupus italicus* [19,20]. The southern part of the Molise region includes the so-called Matese,
which is known for its habitat diversity and birdlife richness [21].

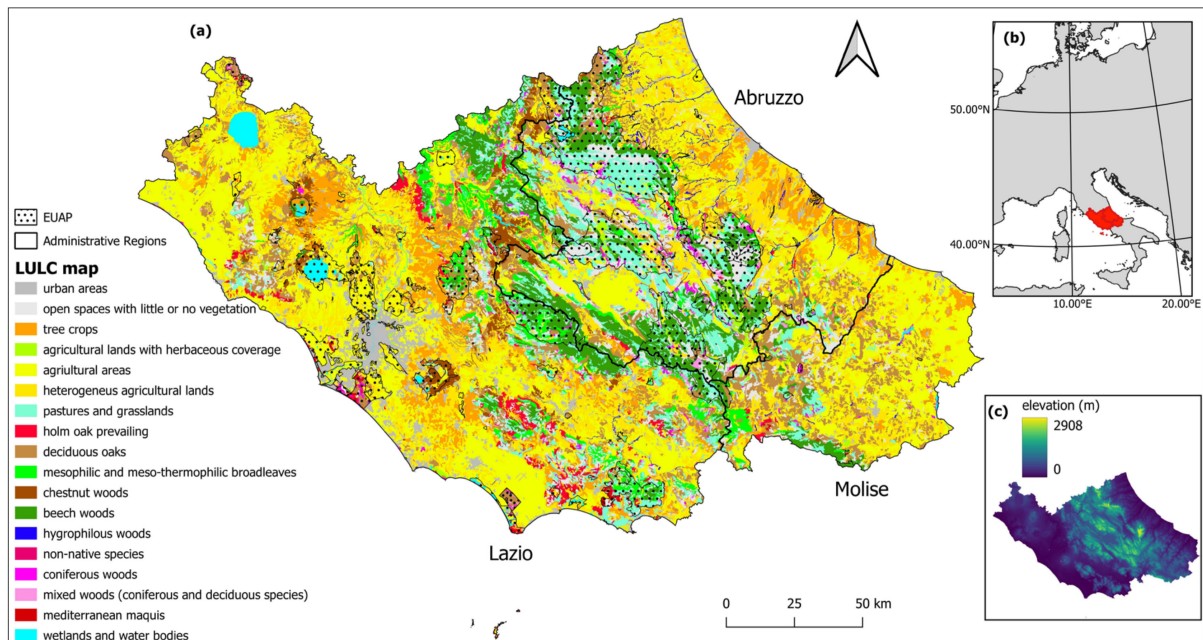

**Figure 1.** Land use and land cover map of the study area (**a**); the administrative regions' borders
are marked in black, and EUAP areas are dotted. The location of the study area (**b**) and the digital
elevation model (**c**) are also shown.

### 2.2. Land Use and Land Cover Map

The LULC map is presented in Figure 1a. The map has been produced by re-arranging
the 2018 CORINE land cover database, which is the primary LULC data source for Europe
and has a thematic accuracy greater than 85% [22]; more detailed information about the
re-arrangement process is provided in Supplementary Materials. For our purposes, we
decided to convey all urban classes into one macro class ("urban areas"); for agricultural
areas, we maintained a separation between crops and tree crops, as the presence/absence
of tree vegetation is going to affect the output of the InVEST models; following Fusaro
et al. (2017) [23], we grouped forests into 10 classes, mostly based on their physiognomic–
structural features. Lastly, water bodies and wetlands were grouped into a single class, and
so were also grassland and pastures.

### 2.3. Ecosystem Services' Spatial Assessment

We assessed three regulating ESs, that is, carbon storage, seasonal water yield, and
urban flood risk mitigation. To do so, we ran three models belonging to the InVEST suite.
InVEST is a free and open-source modeling tool developed by the Natural Capital Project
that uses land-use and land-cover (LULC) patterns to estimate the ES production spatially
and explicitly in biophysical or economic terms [24]. InVEST models have been used all

over the world at both national and regional scales [25–28]; various studies have shown that the models' performances at the regional scale are strictly linked to the quality and resolution of input data [29,30].

The carbon storage was estimated using the carbon storage and sequestration model. This model largely depends on four carbon pools: aboveground biomass, belowground biomass, soil, and dead organic matter; the model aggregates the carbon stored in each of these pools [31]. In our case, the forest carbon pools were estimated according to data provided by the National Inventory of Forests and forests Carbon Pool (INFC) [32], which assesses the carbon pools for 18 forest categories at the administrative region's scale. To ensure the quality of data, the INFC uses a three-phase sampling strategy that also included a field campaign for ground validation [33]. For each forest class, we averaged the values from the three administrative regions covering the study area (Lazio, Abruzzo, Molise). Data in the inventory is reported in biomass; therefore, we converted it into carbon by applying the following equation, which is reported in the National Inventory of Forests and forests Carbon Pool itself:

$$Carbon = 0.5 \times biomass \tag{1}$$

For non-forestry LULC classes, data were collected from the literature, preferably using studies carried out within the study area (see Supplementary Materials). The belowground biomass was computed using root/shoot ratios taken from the literature, which is generally equal to 0.4 (see Supplementary Materials).

The water yield of the study area was assessed using the seasonal water yield model (SWYM). InVEST SWYM allows users to rank pixels of land according to their relative contributions to different components of the hydrological cycle, such as quickflow and baseflow [30]. For our purposes, we used the following outputs of the SWYM:

1.  Local recharge, that is, the amount of precipitation that infiltrates the soil. Local recharge is derived from the local water balance, which is in turn dependent upon precipitation, quickflow, and evapotranspiration.
2.  Baseflow index, that is, the amount of groundwater going into the stream from each pixel and is given by the actual baseflow and the available recharge.

The SWYM requires input data such as curve numbers (CNs), which indicate the runoff potential of each parcel, crop coefficients (kc), which is a measure of the evapotranspiration, climatic zones, digital elevation model (DEM), and precipitation; here, we present the principal data sources.

CNs were obtained from the USDA (United States Department of Agriculture) handbook. The crop coefficient (kc) of forest classes was computed following Equation (2), reported by Sharp et al. (2020):

$$\begin{cases} \frac{LAI}{3} \ when \ LAI \leq 3 \\ 1 \ when \ LAI > 3 \end{cases} \tag{2}$$

The mean annual leaf area index (LAI) was estimated using the Copernicus Global Land Service (CGLS) LAI product, a freely available product with a 300 m spatial resolution and a temporal resolution of 10 days. Climatic zones were derived by re-classifying the ecoregional classification of Italy proposed by Blasi et al. (2018) [34]; three climatic zones were identified: Tyrrenic (2B), Appennine (1C), and Adriatic (2C). We used the DEM provided by ISPRA (Italian Institute for Environmental Protection and Research), with a spatial resolution of 20 m. The SWYM has recently been validated, and a good match was found between observed and modeled values for areas with annual precipitation lower than 1000 mm, which is the case for the present case study [35].

Lastly, we ran the urban flood risk mitigation model (UFRMM), which computes the amount of runoff retained by a pixel during a storm event to model the runoff reduction [31]. The model uses the SCS-CN approach, which does generally well in capturing the ranking between different land uses, even though it introduces high uncertainties [31]. The runoff

retention was computed on a precipitation event of 100 mm, which is very intense for the study area. For more information about the InVEST models, we recommend consulting the model's documentation [31]; all data sources can be found in Supplementary Materials. We then quantified the spatial relations of ESs by computing pairwise Pearson's correlation coefficients; indeed, assessing ESs synergies and tradeoffs is crucial for comprehensive spatial planning of protected areas [36,37].

### 2.4. Identification of Proposed Protected Natural Areas

ESs were assessed using different units; therefore, we re-scaled each ESs map in a range included between 0 and 100. To do so, we used the r.rescale.eq module in GRASS GIS, which allows for rescaling the range of values in a raster map with equalized histogram. Then, for each ES, we produced a map containing pixels falling in the 80th percentile and above, which we called high delivery areas (HDAs); priority areas (PAs) were found through overlay analysis by overlapping HDAs. A similar methodology has been used in several studies, in some cases with slightly different cutoff thresholds [38–40]. To identify the proposed protected areas (PPAs), we went through the following steps. First, we produced a new map of all those PAs which are not currently falling within the EUAP network. Then, using the r.clump module in GRASS GIS, we extracted all patches bigger than 140 km$^2$; this threshold was arbitrarily chosen, as it allows for the identification of extended areas, which are connected in some cases. The resulting patches were then converted into polygons using the r.to.vect module in GRASS GIS.

### 3. Results and Discussion

Figure 2 reports ESs maps and HDAs. Despite slightly different trends, all ESs reach the highest level of production in the central Apennine in mountain forested areas. This is because the considered ESs strongly depends on ecological processes such as photosynthesis, carbon storage, runoff interception, and rainfall retention, which are notoriously favored by the presence of forests rather than urban or agricultural areas [41–43]. Interestingly, due to the absence of woody vegetation, coastal areas, and dry grasslands above the three-line limit also show a limited potential in delivering ESs. However, one should not forget that these areas provide crucial ESs besides the considered ones, such as biodiversity conservation [17,44,45], atmospheric pollution mitigation [4], and outdoor recreation [7,46].

As for the ESs synergies, pairwise Pearson's correlation coefficients are shown in Table 1. The pairwise baseflow-local recharge and carbon stock-runoff retention have a Pearson's correlation coefficient of 0.91 and 0.74 respectively; other ESs pairwise show a Pearson's correlation coefficient between 0.26 and 0.35. Despite being generally considered negligible [47], a Pearson's correlation coefficient above 0.2 has often been considered meaningful in ESs studies [48,49]; therefore, we conclude that the selected ESs can be considered as a bundle, that is, they co-occur repeatedly in landscape and show patterns derived from the LULC types [50]. Identifying ESs bundles could avoid potential trade-offs generated by uninformed policymaking [51]; according to each case, it may allow for policymakers to adopt a land-management strategy tailored for preserving one or more bundles or to enhance the deficient ones. ESs bundles are also potentially useful for the management of protected areas, where regulating, cultural and provisioning services are strongly interlinked, and for assessing the preferences of tourists, which must be taken into account, amongst the others, for the income they generate [51]. Functional zoning based on ESs bundles has therefore the potential to be the nexus between landscape spatial planning and social policies, enabling the definition of win-win management solutions [52,53]. However, for our study area, much more effort by the scientific community is needed, as studies aimed at defining ESs bundles lack.

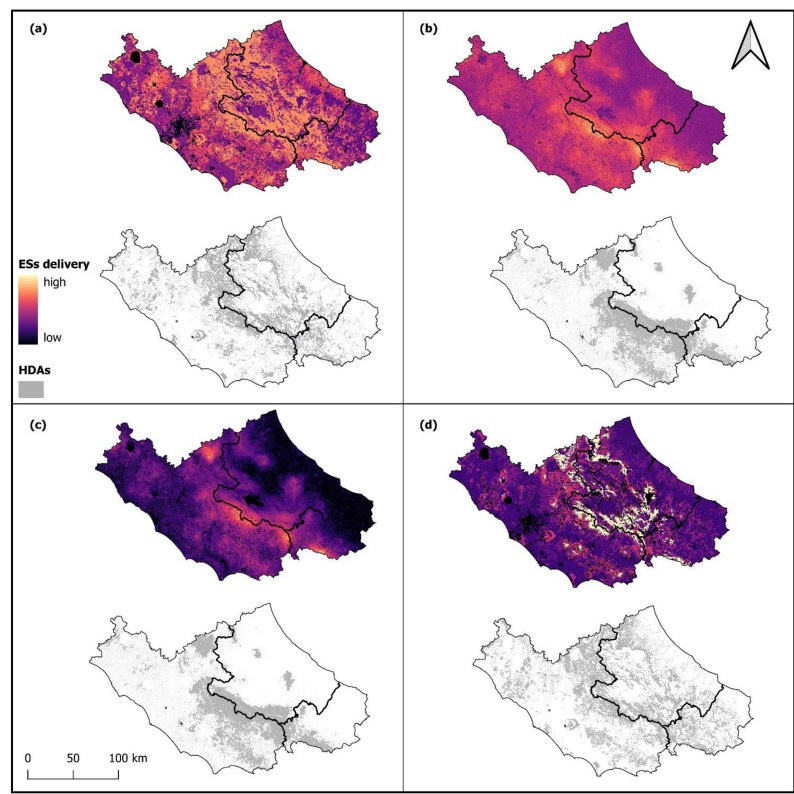

**Figure 2.** Ecosystem services' delivery maps and high delivery areas (HDAs) for runoff retention (**a**), local recharge (**b**), baseflow (**c**), and carbon stock (**d**).

**Table 1.** Pearson's correlation coefficient for pairwise Ecosystem Services. n = 3,240,030.

|  | **Baseflow** | **Local Recharge** | **Runoff Retention** | **Carbon Stock** |
|---|---|---|---|---|
| Baseflow | 1 | / | / | / |
| Local recharge | 0.93 | 1 | / | / |
| Runoff retention | 0.35 | 0.28 | 1 | / |
| Carbon stock | 0.32 | 0.26 | 0.74 | 1 |

Figure 3 shows the priority areas (PAs), alongside the EUAP network and the proposed protected areas. The PAs cover about 2676.9 km$^2$ (8.23% of the study area); only 31.5% of these areas are currently protected by the EUAP network. Therefore, despite the lack of studies for comparison, we believe that there is room for considerable improvement in the PAs' protection.

The PPAs cover 887.9 km$^2$, which equates to 2.73% of the study area (Table 2) and consist of four separate patches that will be discussed later in this section. The combination of EUAP and PPAs covers about 19% of the study area (Table 2); this means that the goal of protecting strictly at least 10% of land by 2030 would be almost doubled if PPAs are eventually established. Plus, with the proposed intervention, the priority areas under EUAP protection would more than double, raising to 1727.5 km$^2$, that is, 64.5% of the total PA (Table 2).

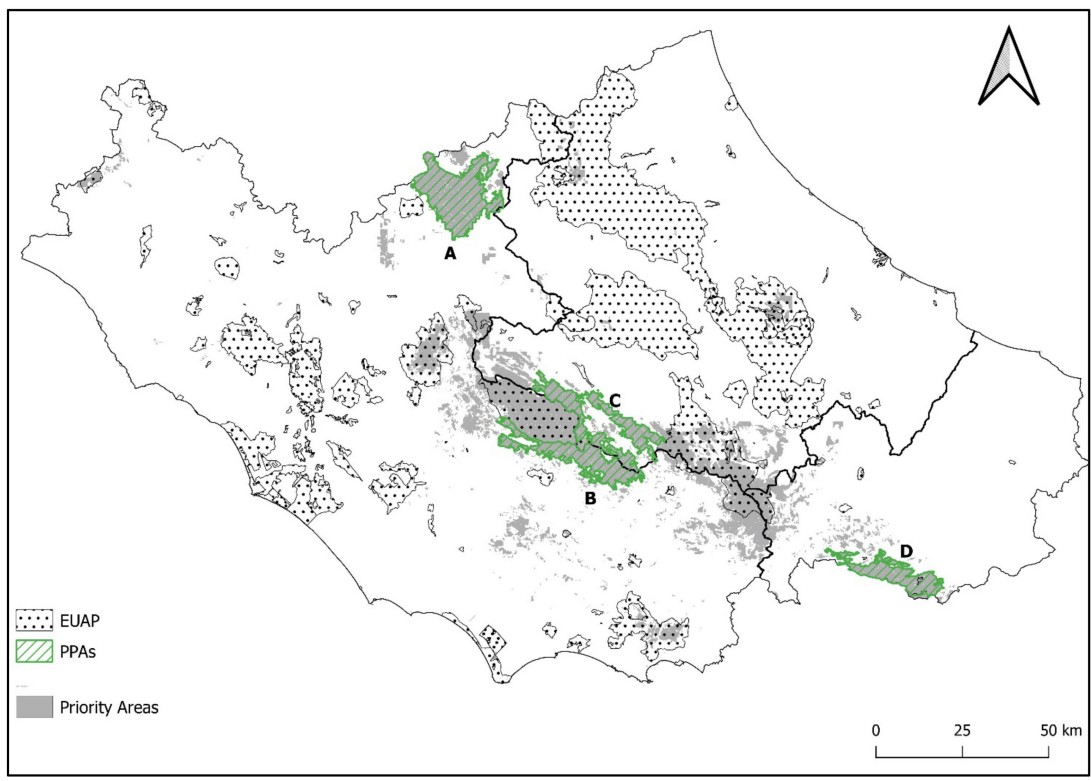

**Figure 3.** EUAP network (dotted), proposed protected areas (PPAs; the four identified patches are indicated by the letters A,B,C and D), and the ecosystem services' priority areas (PA).

**Table 2.** Study area and priority areas' coverage by EUAP, PPAs, and by their combination.

|  | Coverage of the Study Area | | Coverage of PAs | |
|---|---|---|---|---|
|  | km$^2$ | % | km$^2$ | % |
| EUAP | 5277.3 | 16.23 | 842.3 | 31.5 |
| PPAs | 887.9 | 2.73 | 885.2 | 33.1 |
| EUAP + PPAs | 6164.99 | 18.96 | 1727.5 | 64.5 |

Patch A (Figure 3) extends for 316.88 km$^2$ and covers the Monti Reatini, including one of the most important mountains of Region Lazio, Mount Terminillo. Altitude ranges from 600 m a.s.l to about 2200 m a.s.l, with a mean value of 1235 m a.s.l. About 80% of this area is covered by forests; beech wood forests are the most prevalent ones, covering about 56% of the total area. Dry grasslands are also abundant, covering about 13% of the total area. There is a large consensus on the need for better protection of this area [15], which has undergone remarkable touristic exploitation [54] that lead, over the last 50 years, to the construction of several ski facilities [55]; the creation of a new green corridor, connecting the Monti Reatini and the adjacent Gran Sasso and Monti della Laga National Park (Figure 3), has also been hypothesized [15]. To date, the only form of protection resides in six different Natura2000 sites, which cover the majority of the area. Given the ecological continuity of these Natura2000 sites, which share the same scientific, landscape, and cultural values, we argue that they would greatly benefit from integrated, unitary, and comprehensive management, which may result from the institution of a new national protected area, such as a regional park.

Patches B and C, which we will combine as they do not connect for a few pixels, extend for 414.1 km$^2$ and stretch on the so-called Monti Ernici, right at the border of two important EUAP areas, namely, the Monti Simbruini regional park on the west and the Abruzzo, Lazio,

and Molise national parks on the eastern side (Figure 3). The altitude spans from 325 m a.s.l to 2140 m a.s.l, with a mean value of 1218 m a.s.l. About 91% of this area is covered by forests; beech wood forests are the most extended ones, as they cover about 62% of the total area. Patch B also includes the Certosa of Trisulti, an abbey built in roughly 1200 that hosts a state library with about 25,000 volumes, whose historical and cultural heritage has been recognized with its appointment as a national monument [56]. Strong protection of this area has already been hypothesized, as it would create a green corridor between two important EUAP areas and extend the habitat of *Ursus arctos marsicanus* [15,57]; therefore, we argue that strict protection of patches B and C has a notable value in several aspects besides ESs' protection. The majority of the area is within the Natura2000 network, under 14 different sites; therefore, as for Patch A, we conclude that this area would greatly benefit from the institution of a new EUAP area such as an interregional park, which may provide better and more comprehensive management. A legislative proposal (law proposal n.155, 22 May 2019) for the institution of a new EUAP area partially covering Patches B and C is being evaluated, even though, to date, national institutions have shown a certain disinterest, thus slowing down the discussion.

Patch D (Figure 3) extends for 140.48 km$^2$ and lies over the Matese area, with an altitude spanning from 439 to 2032 m a.s.l, and a mean value of 1134 m a.s.l. About 80% of the total area is forested; beech woods are the most extended forest type, covering 47% of the total area of the patch. Patch D is almost entirely covered by one single Natura2000 site. This PPA is also connected to the adjacent Matese Regional Park, which has been instituted in 2002 in Campania region, and it is therefore out of the study area. A national law (n.205, 17 December 2017) for the institution of a national park covering this area has already been approved; therefore, we believe that this PPA will likely turn into a EUAP area within the next few years.

It's noteworthy that the discussion around the institution of new EUAP areas in Central Italy has little, to date, been supported by ESs studies. Therefore, we argue that this research will help stakeholders and policymakers in gaining an ES-based perspective and provide new pieces of evidence on the importance of protecting these areas. It is also worth remarking that, in our case study, the current EUAP network shows a major conflict; indeed, extended mountain forested areas, characterized by high ESs production, as well as a remarkable ecological and cultural value and already within the Natura2000 network, do not fall under any national nature conservation scheme.

This study does not pretend to be exhaustive. Indeed, one must keep in mind that (i) we only considered 3 ESs, whereas many others could have been included in the analysis, and (ii) establishing new EUAP areas is complex and conflicts associated with the social and economic impacts imposed on local communities cannot be neglected [1,58]. A common critical aspect is the one regarding the co-existence of sustainable economic development (which is mainly driven by tourism, agricultural activity, the restaurant industry, and local products) and nature conservation in protected areas. In fact, despite the Italian law on protected areas stipulating that economic and social activities shall be encouraged and promoted, it does not always happen, mostly due to the lack of well established or conflicting management tools [59]. To address this issue, the Italian government has recently established the economic environmental zones (zona economica ambientale, ZEA), which match the national parks and are intended as a further management tool aimed at directing new financing for economic activities operating within parks. Indeed, it is important to remember that a successful land planning strategy should be based on an interdisciplinary approach, which is required to fully exploit the multifunctional role of ESs [60].

## 4. Limitations

The use of the InVEST suite's models has flaws that have been reported in the literature [28,61]. The selected models oversimplify the carbon and water cycle, as well as the runoff reduction by the natural areas. To partially overcome this limitation, we used

local and regional input data if available (see Supplementary Materials), as suggested by previous studies [62,63].

Lastly, there is a certain degree of subjectivity. Indeed, we arbitrarily select the 80th percentile upwards as the threshold for setting ESs' HDAs; likewise, we decided to only consider patches bigger than 140 km$^2$ for identifying PPAs.

## 5. Conclusions

This work supports the spatial planning of new protected areas in central Italy. We argue that our work fits the EU policies, which clearly state that setting new protected areas is crucial to protect and enhance ecosystem services and deal with climate change; therefore, our study might help to reach the EU goal of protecting at least 30% of land by 2030.

Using the InVEST suite, we assessed three regulating ESs, that is, carbon storage, seasonal water yield, and urban flood risk mitigation; using overlay analysis, we mapped four proposed protected areas covering 887.9 km$^2$, specifically targeted at protecting and enhancing the ESs delivery.

We found that the current network of nationally protected areas only covers 31.5% of priority areas for the ESs production; in the scenario in which all proposed protected areas are established, this share would rise to 64.5%.

Interestingly, each one of the proposed protected areas has been yet hypothesized and discussed by associations and stakeholders. However, (i) scientific research to support their institution, and specifically, ESs studies, is lacking; and (ii) besides the one falling in the Matese area (patch D), institutions seem not interested in making appreciable efforts for better conservation of these areas.

We hopefully provide new, scientifically sound, and actionable knowledge that might revive the debate on the institution of new protected areas in central Italy; plus, this methodology may also be replicable at the city scale, supporting the urban reforestation/afforestation initiatives funded by the National Recovery and Resilience Plan.

**Supplementary Materials:** The following supporting information can be downloaded at: https://www.mdpi.com/article/10.3390/f14010145/s1.

**Author Contributions:** Conceptualization, Methodology, Software, Writing—original draft, A.S.; Writing—review and editing, Supervision, Funding acquisition, S.F. All authors have read and agreed to the published version of the manuscript.

**Funding:** Regione Lazio: project n. 36388 TECNOVERDE: "Tecnologie geomatiche e ambientali di precisione per il monitoraggio e la valorizzazione dei servizi ecosistemici delle infra-strutture verdi urbane e peri-urbane"; 2021 @CNR project BIOCITY "Riforestazione urbana: nuovi strumenti conoscitivi e di supporto decisionale"; PRIN2020-MULTIFOR"Multi-scale observations to predict Forest response to pollution and climate change"; CIR01_00019—PRO-ICOS_MED Potenziamento della Rete di Osservazione ICOS-Italia nel Mediterraneo—Rafforzamento del capitale umano" funded by the Ministry of Research.

**Data Availability Statement:** The data and data sources presented in this study are available in Supplementary Materials.

**Conflicts of Interest:** The authors declare no conflict of interest.

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
