# Peer review of "Spatial Prioritization of Ecosystem Services for Land Conservation: The Case Study of Central Italy"

_forests, doi:10.3390/f14010145_

Round 1
Reviewer 1 Report
The paper identifies priority areas for biodiversity conservation and delivery of three ecosystem services (ESs) in central Italy. The values of the ecosystem services are mapped and areas identified for which all three ESs are in their respective upper 20%-quantiles. Finding such joint areas is possible because all three ESs are positively spatially correlated.
The analysis is sound and the results practically relevant. One question I’d like to see discussed a little is what would happen if there are negative correlations between the ESs (which might well appear if other ESs are considered as well). I’d expect some trade-off emerging here, so that one has make decisions like whether it is better to conserve one larger area that is average in all ESs or a number of smaller areas each of which is good in one or few ESs.
As a formal issue, in eq. (2) the condition (“LAI>3” or “otherwise”) for the outcome kc=1 is missing.
Author Response
We would like to thank all of the reviewers for the valuable comments that helped us in improving the overall quality of the manuscript. Our modifications are reported in red in the new version of the manuscript and supplementary materials.
We have better discussed the aspect concerning the ESs bundles in lines 222-229. Despite we didn’t explicitly tackle the aspects regarding smaller vs larger areas, we touched relevant aspects such as the ESs tradeoffs and the potential application of ESs bundles for a better management of protected areas.
The formal issue in equation 2 has been fixed.
Reviewer 2 Report
I have some concerns and comments regarding your submission.
1. The title is not appropriate, actually your study does not involve spatial optimization.
2. Please clearly declare how did you optimize the landscape configuration, and what are the potential applications for land management in both conclusion and abstract sections.
3. The approach you applied (i.e., overlay analysis) is too simple to identify the so called protected areas, more indicators and approaches should be applied to integrate ESs into landscape management.
4. Line 24: How to logically link your research findings with the goal of protecting at least 30 % of land in Europe by 2030.
5. I do not think three ESs (e.g., carbon storage, seasonal water yield, and urban flood risk mitigation) are sufficient to reflect ecosystem states, habitat maintenance and soil retention should be included via the tool of InVEST.
6. More technical details are needed to introduce the input parameters of InVEST model.
7. How to validate and calibrate the estimate results of ESs.
8. How to validate the LUC classification maps.
Author Response
We would like to thank all of the reviewers for the valuable comments that helped us in improving the overall quality of the manuscript. Our modifications are reported in red in the new version of the manuscript and supplementary materials.
- The title is not appropriate, actually your study does not involve spatial optimization.
The title is the following: “Spatial prioritization of ecosystem services for land conservation: the case study of Central Italy”. in this study, we identified priority areas for the provision of ecosystem services. That said, we do not deal with optimization, that is, “The process of finding the best possible solution to a problem.” (from the Oxford dictionary). Therefore, since we actually prioritized central Italy according to the provision of ESs, we think the title is appropriate.
- Please clearly declare how did you optimize the landscape configuration, and what are the potential applications for land management in both conclusion and abstract sections.
Here I quote from the abstract (lines 17-22) : “[…] we found areas with the highest delivery in each of the considered ESs; based on these findings, we eventually proposed 4 new protected areas, which combine for 888 km2, that is, 2.73 % of the study area. Interestingly, each of the newly proposed protected areas has somehow been discussed and hypothesized by stakeholders, but only one is presumably going to be part of the national network of protected areas within the next years.”. Here I quote from the conclusion section: “[…] We found that the current network of nationally protected areas only covers 31.5% of Priority Areas for the ESs production; in the scenario in which all Proposed Protected Areas are established, this share would rise to 64.5%. Interestingly, each one of the Proposed Protected Areas has been yet hypothesized and discussed by associations and stakeholders. However, i) scientific research to support their institution, and specifically, ESs studies, lack; and ii) besides the one falling in the Matese area (patch D), Institutions seem not interested in making appreciable efforts for better conservation of these areas.“
In our opinion, these are clear practical applications for land management coming out from our study, as we are suggesting a stronger regime of protection in these specific areas.
- The approach you applied (i.e., overlay analysis) is too simple to identify the so called protected areas, more indicators and approaches should be applied to integrate ESs into landscape management.
Despite being simple, this or similar approaches have been widely used in many studies published in top-ranked journals. Please see:
- Mokondoko, P., Manson, R. H., Ricketts, T. H., & Geissert, D. (2018). Spatial analysis of ecosystem service relationships to improve targeting of payments for hydrological services. PloS one, 13(2), e0192560.;
- Qiu, J., & Turner, M. G. (2013). Spatial interactions among ecosystem services in an urbanizing agricultural watershed. Proceedings of the National Academy of Sciences, 110(29), 12149-12154.;
- Orsi, F., Ciolli, M., Primmer, E., Varumo, L., & Geneletti, D. (2020). Mapping hotspots and bundles of forest ecosystem services across the European Union. Land Use Policy, 99, 104840.).
Therefore, we believe that it’s a good starting point for integrating ESs into land management. Plus, one should not forget that InVEST models are based on parameters such as Leaf Area Index (that is, the amount of foliage), kc (crop factor, that is, the water requirement) as well as land use and land cover data, which are indeed solid and reliable indicators for landscape management.
- Line 24: How to logically link your research findings with the goal of protecting at least 30 % of land in Europe by 2030.
In line 22 we specified that the logical link is the prioritization of areas according to their production of ESs.
- I do not think three ESs (e.g., carbon storage, seasonal water yield, and urban flood risk mitigation) are sufficient to reflect ecosystem states, habitat maintenance and soil retention should be included via the tool of InVEST.
As we have stated throughout the manuscript, this study does not pretend to be exhaustive; indeed, it is the first step toward a comprehensive evaluation of ecosystem services in Central Italy. Therefore, despite habitat maintenance and soil retention being two pivotal ecosystem services provided by forests, we think that those would not add as much to the discussion. We thank the referee for the valuable comments, which we will consider for our future work.
- More technical details are needed to introduce the input parameters of InVEST model.
We added more technical information concerning the input data both in the manuscript, in order to make our workflow and methodology clearer. Plus, we added some details concerning the model’s functioning. The modifications have been made in the Materials and methods section. We thank the referee for helping us in improving this section.
Please note that we would like to keep most of this information in the supplementary materials; indeed, the aim of this manuscript is not to illustrate technical aspects of the model’s functioning.
- How to validate and calibrate the estimate results of ESs.
Validating and calibrating the ESs estimates is beyond the goal of this study. However, in this new version, we remarked that:
For carbon storage, the input data were collected according to the methodology described by Fattorini et al. (2006). The Authors adopted a three-phase sampling strategy that also included a field campaign for the ground validation. Please find more in the following paper: Fattorini, Marcheselli M., Pisani C. (2006) A three-phase sampling strategy for large-scale multiresource forest inventories. Journal of Agricultural, Biological and Environmental Statistics, 11: 296-316. Modifications to the text have been made at lines 140-142.
InVEST SWYM has been recently validated by Benra et al. (2021), who found that the model works pretty well for basins with less than 1000 mm of precipitation/year, as in our case. Please find more in the following paper: Benra, , De Frutos, A., Gaglio, M., Álvarez-Garretón, C., Felipe-Lucia, M., & Bonn, A. (2021). Mapping water ecosystem services: Evaluating InVEST model predictions in data scarce regions. Environmental Modelling & Software, 138, 104982.
For the runoff retention, we used the same parametrization (including some of the input data) as other studies (Kadaverugu, A., Kadaverugu, R., Chintala, N. R., & Gorthi, K. V. (2022). Flood vulnerability assessment of urban micro-watersheds using multi-criteria decision making and InVEST model: a case of Hyderabad City, India. Modeling Earth Systems and Environment, 8(3), 3447-3459.).
Please see lines 139-143; 152; 155-156 of the new version
- How to validate the LUC classification maps.
The LULC map was retrieved using the Corine Land Cover dataset. The Corine Land Cover is the most widely used land cover dataset in Europe and has been validated by independent experts who evaluated its accuracy using more than 25.000 sampling locations. Therefore, we argue that the validation of the LULC data is not needed for our study. Please find more information at the following link: https://land.copernicus.eu/faq/clc-1.
Please see lines 118-119 of the new version
Round 2
Reviewer 2 Report
1. The land use/cover maps should be validated for regional study, although it has been widely applied in surrounding areas.
2. The aim of this manuscript is not to illustrate technical aspects of the model’s functioning, BUT you have to convince me the estimated results are correct, and the comparison and validation are needed.
Author Response
- The land use/cover maps should be validated for regional study, although it has been widely applied in surrounding areas.
The Corine Land Cover is rigorously validated at a 3-stage level, that is, pan-European, country (or group of countries), and biogeographical region. Overall, the thematic accuracy assessment includes more than 25,000 points, selected in a blind approach using the LUCAS sampling frame.
As you can read in table 8 (page 38) Italy shows a plausibility accuracy level above 85 %, which is remarkable. This result has been reached by checking almost 1300 ground points (page 74).
Additional validation has been performed for Mediterranean areas, including Spain, Greece, and central-southern Italy. Also in this case the plausibility accuracy is fairly above 85%, with almost 4400 ground points (page 121). Here I send you the link where you can access all the information: https://land.copernicus.eu/user-corner/technical-library/clc-2012-validation-report-1.
For this reason, since the early 90’s, the Corine Land Cover has widely been accepted as a scientifically robust LULC data source. Here I send you some research published in top-ranked journals (including Forests) which used CLC data without providing their own validation:
- Gaglio, M., Aschonitis, V., Castaldelli, G., & Fano, E. A. (2020). Land use intensification rather than land cover change affects regulating services in the mountainous Adige river basin (Italy). Ecosystem Services, 45, 101158.
- Burkhard, B., Kroll, F., Nedkov, S., & Müller, F. (2012). Mapping ecosystem service supply, demand and budgets. Ecological indicators, 21, 17-29.
- Cabral, P., Feger, C., Levrel, H., Chambolle, M., & Basque, D. (2016). Assessing the impact of land-cover changes on ecosystem services: A first step toward integrative planning in Bordeaux, France. Ecosystem Services, 22, 318-327.
- Páscoa, P., Gouveia, C. M., & Kurz-Besson, C. (2020). A simple method to identify potential groundwater-dependent vegetation using NDVI MODIS. Forests, 11(2), 147.
We would like to remark that we understand your point, since validating input data is crucial for providing quality outputs; however, considering that the quality of CLC data has been widely proven all over Europe, we think that our validation would not add anything meaningful to this research.
- The aim of this manuscript is not to illustrate technical aspects of the model’s functioning, BUT you have to convince me the estimated results are correct, and the comparison and validation are needed.
We would like to clarify some aspects concerning our methodological approach.
First of all, our ultimate goal is to produce a good estimate of the provision of ESs, that could be used by stakeholders and policymakers for management purposes. For this reason, we did not provide results in biophysical terms (e.g carbon stock in Mg ha-1, local recharge, and baseflow in mm), and rather converted these to a value between 0 and 100. That said, we put a massive effort in properly parametrizing the model. We used in many cases strictly local datasets (see the National Forestry Inventory, which is compiled ad the administrative’s region level; the precipitation and evapotranspiration data, produced for Italy by the Italian Institute for Environmental Protection and Research; the climatic zones distribution, provided by the National Institute of Statistics), as recommended by the official InVEST documentation and several other research.
We totally agree that is always better validating models against other independent data (which we will probably do in a forthcoming work). For this reason, InVEST models have widely been validated all over the world. Here I show you some relevant references:
- Benra, F., De Frutos, A., Gaglio, M., Álvarez-Garretón, C., Felipe-Lucia, M., & Bonn, A. (2021). Mapping water ecosystem services: Evaluating InVEST model predictions in data scarce regions. Environmental Modelling & Software, 138, 104982.
- Pache, R. G., Abrudan, I. V., & Niță, M. D. (2020). Economic valuation of carbon storage and sequestration in Retezat National Park, Romania. Forests, 12(1), 43.
- Redhead, J. W., Stratford, C., Sharps, K., Jones, L., Ziv, G., Clarke, D., ... & Bullock, J. M. (2016). Empirical validation of the InVEST water yield ecosystem service model at a national scale. Science of the Total Environment, 569, 1418-1426.
- Redhead JW, May L, Oliver TH, Hamel P, Sharp R, Bullock JM. 2018. National scale evaluation of the InVEST nutrient retention model in the United Kingdom. Sci Total Environ 610–611:666–77.
Therefore, most of scholars use InVEST relying on previous validation studies, just like we did; indeed, based on what found by Ochoa et al (Ochoa, V., & Urbina-Cardona, N. (2017). Tools for spatially modeling ecosystem services: Publication trends, conceptual reflections and future challenges. Ecosystem Services, 26, 155-169.), only 12.9% of InVEST users adopt some kind of validation. This is because validating the performance of a model is most of the time a technical and entirely dedicated research. Indeed, once the validation is performed, one must address why field and modelled data are different, what are the possible improvements to the model, compare his validation to other ones and so on. Despite being extremely interesting, all that work would divert the attention from the main focus of the work, that is, providing policy guidance for land conservation.